# Comparison of CW NUV and Pulse NIR Laser Influence on PbSe Films Photosensitivity

**Anastasiia A. Olkhova** [1,*]**, Alina A. Patrikeeva** [1]**, Maria A. Dubkova** [1]**, Natalia K. Kuzmenko** [2]**, Nikolai V. Nikonorov** [2] **and Maksim M. Sergeev** [1]

1 Institute of Laser Technologies, National Research University ITMO, St. Petersburg 197101, Russia
2 Faculty of Photonics, National Research University ITMO, St. Petersburg 197101, Russia
* Correspondence: anastasiia.olkhova@gmail.com

**Abstract:** This paper shows the laser irradiation effect on optical characteristics on PbSe chalcogenide films as a result of irreversible structural modification. The features of film structure and property modification under the action of a continuous wave (CW) laser with a 405 nm wavelength and nanosecond laser pulses with a 1064 nm wavelength are studied. The valence and conduction bands boundaries displacement as a laser radiation result of photothermal action on the film until it darkens and bleaches, is demonstrated. Under CW near ultraviolet (NUV) laser action, the film was modified at a power density of 0.74 to 1.09 kW/cm$^2$. The near-infrared (NIR) laser pulses used ensured the film structure modification at a power density of 1.45 kW/cm$^2$ and a pulse duration of 4 to 20 ns. Scanning with a laser spot in these modes provides the desired change in the film's optical characteristics, and this becomes a serious alternative to the technology of heat treatment in an oven.

**Keywords:** laser modification; PbSe films; optical characteristics; darkening mode; bleaching mode; heat treatment; continuous wave; laser pulses





## 1. Introduction

PbSe chalcogenide films are widely used as photosensitive elements in gas analysis devices, where high absorption in the IR spectrum from 1 to 4 μm [1–4] and low electrical resistance are important characteristics. Gas analysis devices are important for monitoring the environment and protecting it from emissions of various poisonous gases into the atmosphere. The use of these devices is a prerequisite for the safe operation of large oil refining, chemical and other enterprises, the production of which is associated with toxic substances that are harmful to nature and human health [5]. In addition, research and modification of the photosensitive materials' optical and electrical properties are becoming increasingly popular [6]. Changing optical and electrical properties and tuning them within small ranges of values is a very promising task for many applications. For example, optoelectronic systems, solar cells, and LED technologies [7–9] can be used as photoelectric sensors [10]. The optical and electrical properties of PbSe films are largely affected by the oxidation process, which in most technological processes is activated during heat treatment in an oven. This method of increasing the PbSe film's photosensitivity is technologically difficult to implement, which, in most cases, cannot be controlled and leads to a high percentage of defects at the photodetectors' manufacturing stage.

An alternative to heat treatment in an oven is structural and property laser modification of such films [11,12]. The use of laser irradiation makes it possible to carry out a local structure modification and, at the same time, a predictable change in the optical and electrical film characteristics. The film photothermal effect during laser irradiation leads to sharp and local material heating, followed by structural phase modification due to high temperature (gradient) and heating/cooling rate [13]. This structural modification results in photosensitivity laser correction of the material in a certain spectral range [14,15].

The development and implementation of this technology will significantly improve the sensitivity of gas analysis devices as well as change their design, making them more compact and less energy-consuming. In addition, laser processing of chalcogenide films, in contrast to heat treatment in an oven, can be fully automated, which will increase the yield of suitable products and reduce production costs. An equally important task is to evaluate the chalcogenide film quality after laser irradiation, taking into account changes in the material's optical and electrical characteristics. The solution to problem solution will make it possible to improve the functional elements of optoelectronics, photovoltaics, sensorics, and microanalytics [16–20].

In the present work, the change in the sample's optical characteristics was carried out by modifying the structure of PbSe chalcogenide films with continuous laser radiation in the near UV range and pulsed IR radiation.

## 2. Materials and Methods

The films under study were created by vacuum-thermal sputtering on a plane-parallel cover glass substrate 0.2 mm thick. The film samples were manufactured by OOO Optosens (Saint-Petersburg, Russia). In addition to the initial samples, in the experimental part, films were used after heat treatment in an open furnace under atmospheric conditions. Thermal treatment lasted about 90 s and consisted of heating the film to a temperature of 540 °C, followed by exposure and further increase to a temperature of 630 °C. This treatment included the processes of sample activation and sensitization and was carried out by OOO Optosens [21].

The original films that underwent laser modification were not subjected to heat treatment before. This study is aimed at comparing several processing modes of the original film (heat treatment, laser modification with different radiation sources) in order to identify the most photosensitive samples.

### 2.1. Laser Modification

The chalcogenide film laser modification was carried out by continuous radiation of an LSR405CP-1W semiconductor laser with a wavelength of 405 nm as well as by nanosecond pulses of a fiber laser based on the Minimarker-2 complex (OOO Laser Center, Saint-Petersburg, Russia) with a wavelength of 1064 nm.

In the mode of irradiation with continuous radiation of a semiconductor laser, the central part of the laser spot was diaphragmed, after which the radiation was focused by a lens with a magnification of $20\times$ and a numerical aperture of NA = 0.65. The sample with the film was fixed on a three-coordinate table, Thorlabs MTS50/M-Z8 (Figure 1), and processing was carried out in the focal plane of the lens. An ocular camera was used to visualize the irradiation zone, and the sample was moved into the image plane of the lens. The power density of the laser radiation acting on the film ranged from 0.74 to 1.09 kW/cm$^2$, with a spot diameter of about 30 µm. The intensity profile in the cross-section of the laser beam hitting the film was flat on top (top-flat). The film structure was modified at a laser spot scanning speed of 1 mm/s.

To modify the film structure with NIR laser pulses, a two-mirror galvanometric scanner was used to ensure spot motion along a given trajectory. After the scanner mirrors, the laser beam was focused by an F-theta lens with a processing field of $100 \times 100$ mm (Figure 2). The films were processed in the F-theta lens focal plane. The maximum power density of the laser radiation incident on the film in a laser spot with a diameter of about 50 µm reached 1.45 kW/cm$^2$ with a pulse duration of 4 ns to 20 ns and a pulse repetition rate of 5 kHz to 50 kHz. The laser spot velocity varied from 13 to 120 µm/s.

### 2.2. Optical Measurement

A preliminary analysis of the PbSe films' optical properties before and after exposure to radiation from various laser sources, as well as after thermal treatment, was carried out

using optical microscopy in transmitted and reflected light in bright and dark fields. For this, a Carl Zeiss Axio Imager microscope (Germany) was used.

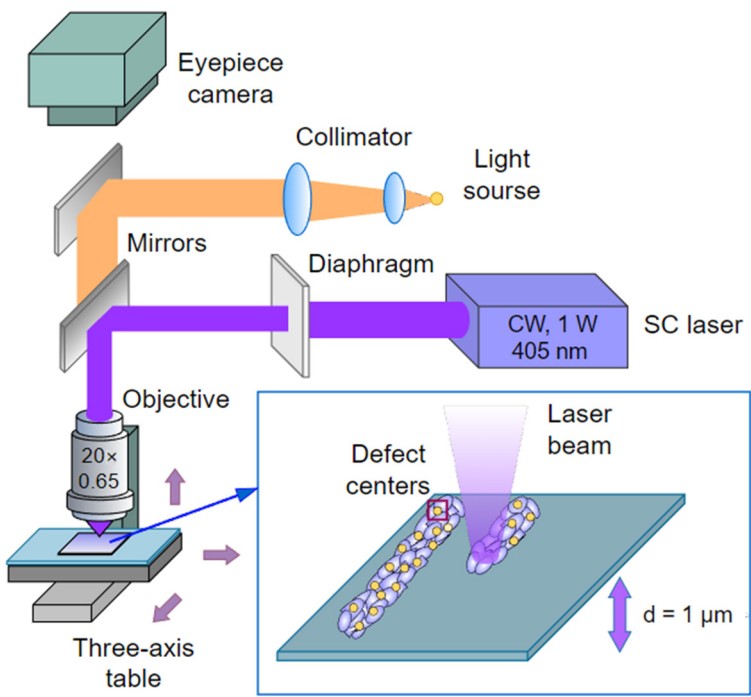

**Figure 1.** Diagram of an experimental setup scheme for laser modification of the structure of chalcogenide films using continuous radiation.

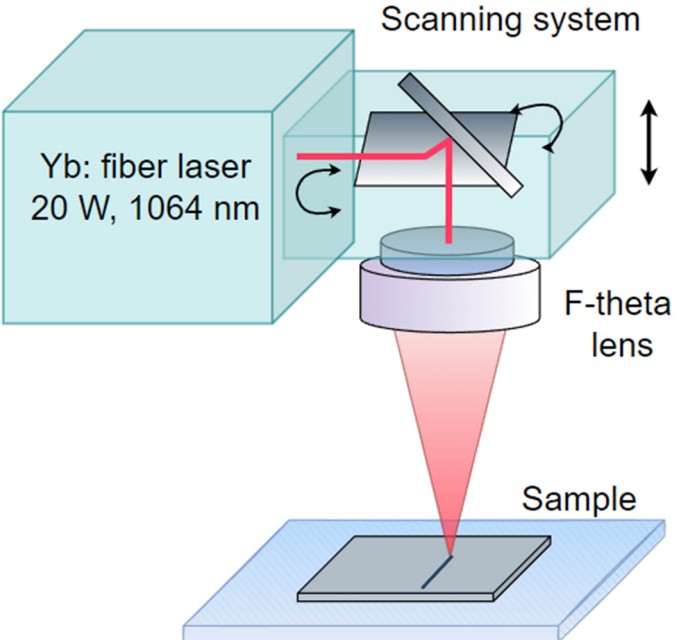

**Figure 2.** Scheme of the experimental setup for chalcogenide film structure laser modification using pulsed radiation.

The reflection and transmission of the samples were measured in the spectral range from 380 to 900 nm using a microscope-spectrophotometer, MSFU-K Yu-30.54.072, LOMO (St. Petersburg, Russia), with a minimum size of the photometric region of 2.0 μm.

To assess the PbSe film's surface using transmission electron microscopy, as well as for elemental sample analysis using energy dispersive X-ray spectroscopy (EDX), a high-

resolution Merlin Zeiss scanning electron microscope (Germany) was used. The reflectance of all samples was measured under normal incidence light.

To analyze the structure of the samples before and after thermal and laser treatment, X-ray diffraction analysis of the films (XRD) was performed using an Ultima IV X-ray diffractometer (Rigaku, Tokyo, Japan). The ICDD PDF2 database was used to identify the phase.

Samples' quantitative analysis and quality control were carried out using an Alpha infrared Fourier spectrometer (Bruker, Bremen, Germany); PbSe films were studied both in the transmission and reflection modes.

## 3. Results and Discussion

### 3.1. Optical Characterization

At the beginning of the study, an analysis of the initial PbSe film structure and samples subjected to heat treatment in an oven was carried out, and their optical characterization is also given (Figure 3).

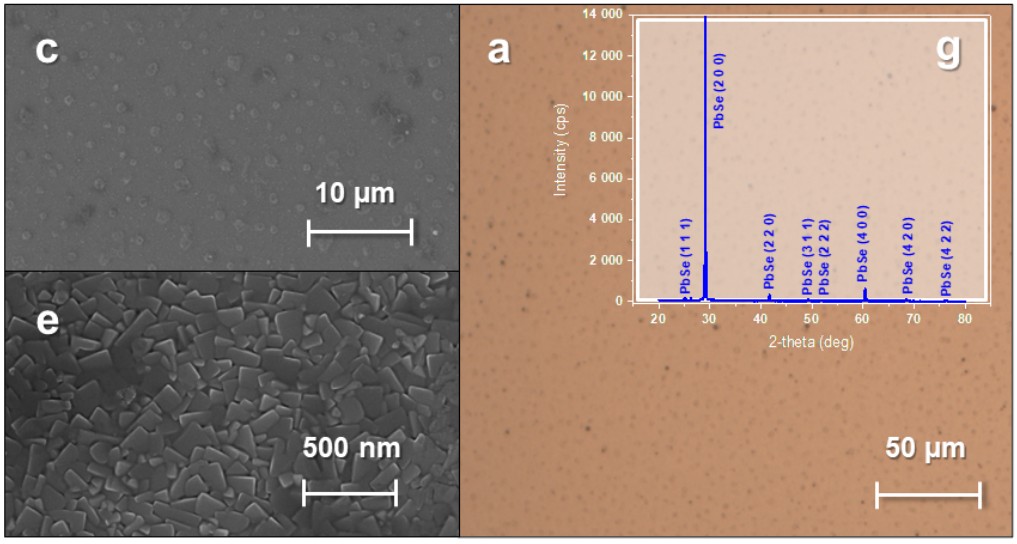

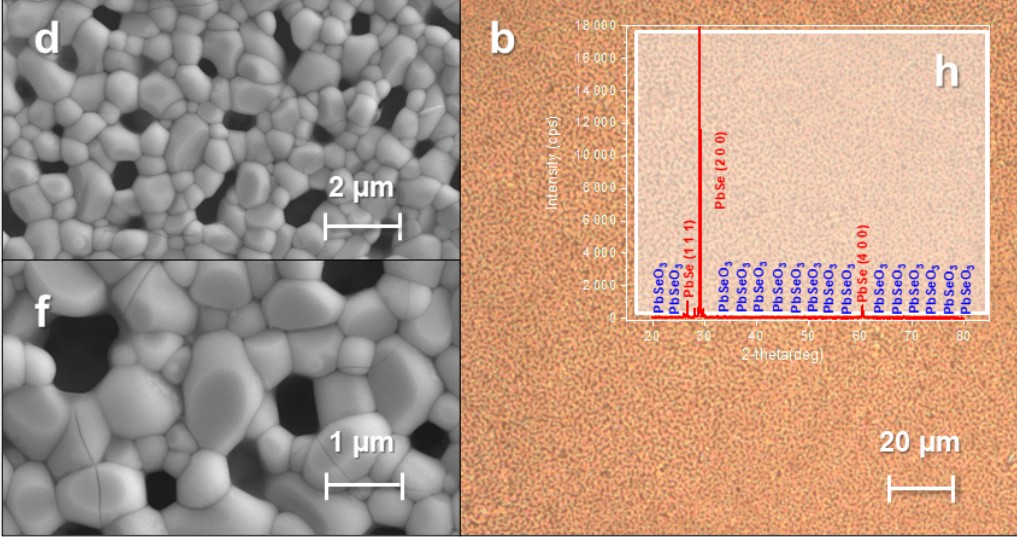

**Figure 3.** Images of a PbSe film obtained using optical microscopy in bright field reflected light: (**a**) initial film; (**b**) film after heat treatment. Pictures of the PbSe film obtained with the SEM secondary electron detector: (**c**,**e**) initial film; (**d**,**f**) film after heat treatment. A picture of the PbSe film obtained using an X-ray diffractometer (**g**) initial film; (**h**) after heat treatment.

During the film's heat treatment in the oven, the inhomogeneities of its structure became the centers of the second phase nucleation ($PbSeO_3$), while the material matrix became denser, which led to a decrease in its transmission. Since the oxide is transparent, an increase in the second phase concentration during heating leads to film bleaching. Moreover, the thermal effect contributes to an increase in the concentration of defect centers, which become secondary sources of free charge carriers in the film, leading to an increase in the conductivity of chalcogenide films. The second phase of high concentration provides a significant increase in the entire film transmission (approximately 30 times). The reflection of the second phase is less, which provides a decrease in the entire sample reflection.

An X-ray of a PbSe film that has undergone heat treatment in an oven (Figure 3h) demonstrates the presence of two phases: PbSe and $PbSeO_3$, whose crystal sizes are 60 nm and 68 nm, respectively. At the same time, for the raw film, the size of the PbSe crystal phase was about 52 nm (Figure 3g).

When films are irradiated, the maximum effects from laser exposure will be observed when the radiation wavelength is chosen close to the film's intrinsic absorption edge. The wavelengths of 405 nm and 1064 nm used in the study fall within the intrinsic absorption region of PbSe films, which provides high absorption of the incident radiation in the absence of transmission.

The most common photoinduced optical changes in chalcogenide films are darkening and bleaching. Upon bleaching, the transmission edge shifts towards higher energies (blue shift), and a corresponding decrease in the absorption coefficient occurs with an increase in the band gap. With darkening, a shift of the transmission edge to lower energies (red shift) is observed, while the absorption increases. The authors of [22] suggest that the observed changes in the optical properties of As-Se-Te are caused by the homopolarization of chemical bonds, namely, an increase in the proportion of As-As, Se-Se, and Te-Te bonds over As-Se and As-Te bonds. As a result, the band gap parameters change, which leads to a shift in the absorption edge. Such a rearrangement of chemical bonds leads to a change in the material polarizability and, consequently, to a change in the permittivity and refractive index. In this case, the material remains amorphous. The effect opposite to darkening—bleaching—is observed in the As-Se system. It leads to an absorption edge shift to the short-wavelength spectrum region and is associated with chemical bond heteropolarization. This effect was detected in the As40Se44Te16 film by increasing the annealing temperature and time. In combination with the darkening effect, local film heating can occur due to the laser radiation's absorption. If the film heating in the affected area reaches the softening or melting temperatures, both a structure change and a redistribution of chemical bonds in the film are possible, which will also lead to changes in the sample's optical properties.

In the present study, the structure of PbSe chalcogenide films was modified using CW and pulsed laser radiation in the focused beam scanning mode. Laser treatment contributed to a change in the optical properties of the samples.

The parameters of laser processing, in which the effects of bleaching and darkening were observed, are shown in Tables 1 and 2. First, a processing mode was discovered in which the material is bleached with subsequent destruction (Figure 4). This mode can be observed when the scanning speed is reduced to 17 mm/s, which leads to an increase in acting energy. With a subsequent decrease in the scanning rate, the PbSe film begins to break down, starting from the track's central part to its edges, which is associated with the radiation intensity's Gaussian distribution over the spot.

**Table 1.** Scanning modes with continuous NUV radiation.

|  | Darkening | Bleaching |
| --- | --- | --- |
| Scanning speed, mm/s | 1 | 1 |
| Power density, $kW/cm^2$ | 0.74 | 1.09 |

**Table 2.** Scanning modes with pulsed NIR radiation.

|  | Darkening | Bleaching |
|---|---|---|
| Scanning speed, mm/s | 120 | 13 |
| Pulse frequency, kHz | 5 | 50 |
| Pulse duration, ns | 4 | 20 |
| Power density, kW/cm$^2$ | 1.45 | 0.39 |

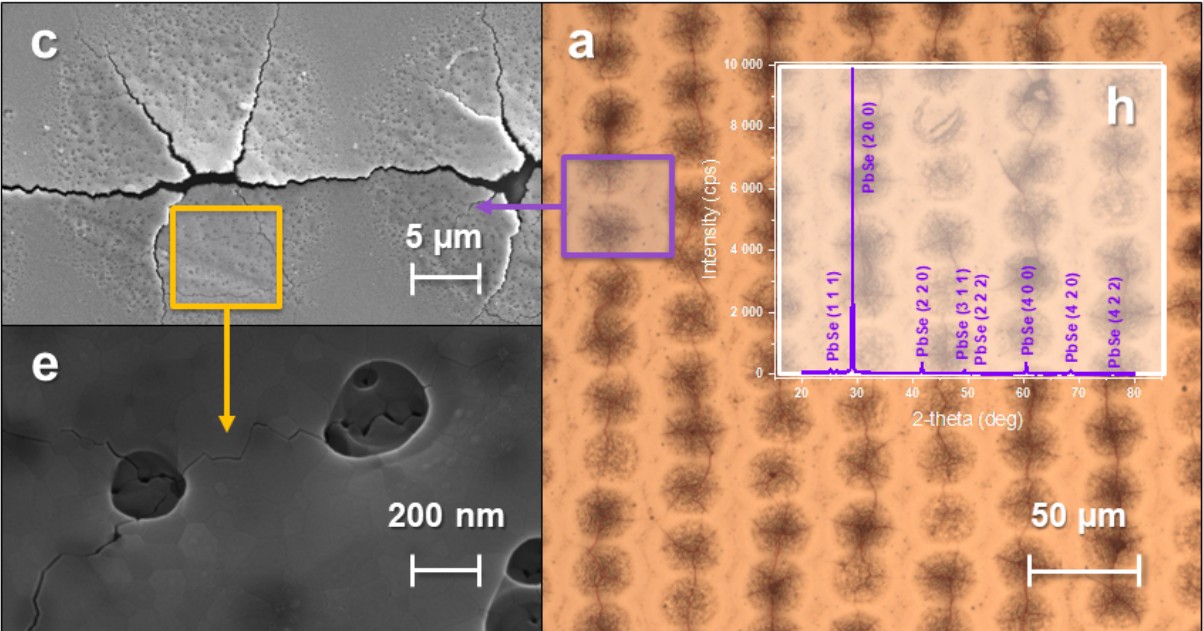

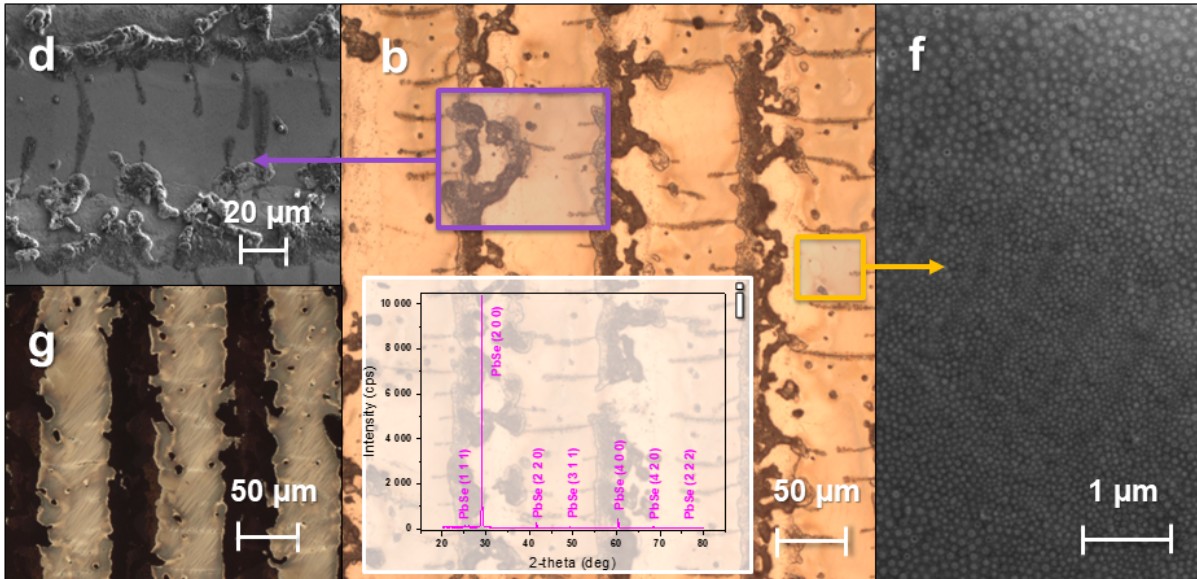

**Figure 4.** Images of a PbSe film obtained using optical microscopy: in a bright field of reflected light, (**a**) darkening mode when modified by pulsed radiation, (**b**) bleaching mode when modified by pulsed radiation; in the transmitted light field (**g**), the bleaching regime upon modification by pulsed radiation. Pictures of the PbSe film obtained with the SEM secondary electron detector: (**c**,**e**) darkening mode when modified by pulsed radiation, (**d**,**f**) bleaching mode when modified by pulsed radiation. Pictures of the PbSe film obtained using an X-ray diffractometer: (**h**) darkening mode when modified by pulsed radiation, (**i**) bleaching mode when modified by pulsed radiation.

The PbSe film's sharp heating due to short pulses (<20 ns) exposure led to the occurrence of excess stress, which caused the crack formation in the laser exposure areas (Figure 4c). When the sample is modified, the material is redistributed, and part of it shifts towards the laser spot center with a moving local heat source and a high temperature gradient. As a result of the samples' treatment in the bleaching mode, the absorption coefficient decreases with increasing band gap, which can be observed in the case of treatment with short pulses and in the authors' previous work [23] for the PbSe films modified by continuous radiation with a wavelength = 405 nm. Additionally, the sample bleaching is characterized by a transmission edge shift towards higher energies in the IR range. With darkening, an increase in absorption and a transmission edge shift towards lower energies are observed. Due to the absorption of laser radiation, the material is heated to a softening state, while the sample structure and the band gap undergo changes, the chemical bonds in the sample are redistributed, and the band gap edge shifts.

After exposure to pulsed laser radiation on the film in the darkening mode, a decrease in the crystal size to 37 nm for the PbSe phase can be observed in the X-ray image (Figure 4h) compared with the raw film. However, after modification of the sample by pulsed radiation in the bleaching mode (Figure 4i), the PbSe phase had crystal sizes of about 65 nm.

Optical characterization was carried out, and the PbSe film structure subjected to laser treatment in the darkening mode was studied. As a result of the laser exposure, periodic structures were formed in the track's central part, which were local hills, since during the melt pool formation, material was periodically drawn into the heat source center, limited by the laser spot. The sample modification did not entail the damage formation in the scanning area and also did not change the original film properties located between the tracks, but it caused melting and contributed to the molten material redistribution. In the laser exposure zone, one can observe the presence of dark and light ridges, which are characterized by different material densities. More dense areas—light ridges—are formed due to the material redistribution and the neighboring dark area depletion of the track, respectively. As in the case of heat treatment in a furnace, a crystal structure is formed at the track periphery due to laser action. However, when laser radiation is used, the formation of an oxide phase is absent. This feature is an advantage since the presence of oxide can contribute to the film's degradation and reduce its service life. Moreover, when the film was rescanned in the mode in which darkening was observed, the film was clarified (Figure 5b).

After exposure to continuous laser radiation on the film both in the darkening mode (Figure 5h) and in the bleaching mode (Figure 5i), an increase in crystal size to 63 nm and 72 nm, respectively, for the PbSe phase can be observed on an X-ray relative to the raw film.

The FESEM image of two films before and after heat treatment is shown in Figure 6a,b. The pictures clearly show the formation of oxide after heat treatment. The presence of constituent elements, such as O, Se, and Pb, in two films is shown in Figure 6a,b.

FESEM images modified with pulsed (Figure 7) and continuous radiation (Figure 8) in darkening and bleaching modes were studied. The composition of the films changes drastically depending on the laser irradiation mode. In the darkening mode, no oxide is formed on the surface (Figures 7a and 8a), which is an important difference in comparison with heat treatment. This may be due to a high temperature gradient and a short laser exposure time. In the bleaching mode, the presence of oxide is observed (Figures 7b and 8b), as well as elements characteristic of the cover glass substrate.

Chemical elements, except Pb, Se, and in some cases O, that is, elements such as Si, Na, Ca, etc. (Figures 7 and 8) belong to the elementary composition of the cover glass—the film substrate. The reason for the detection of these elements is their destruction, including the cracking of the film under the action of laser radiation and the further removal of these elements to the surface of the samples.

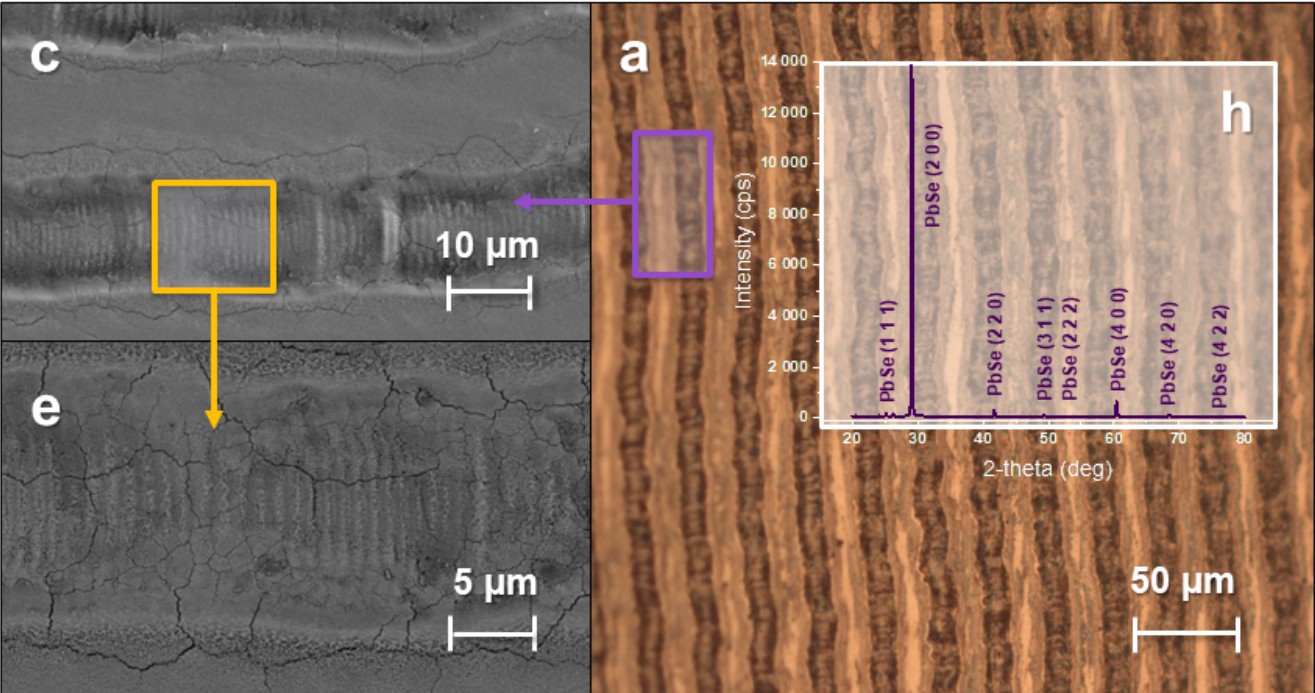

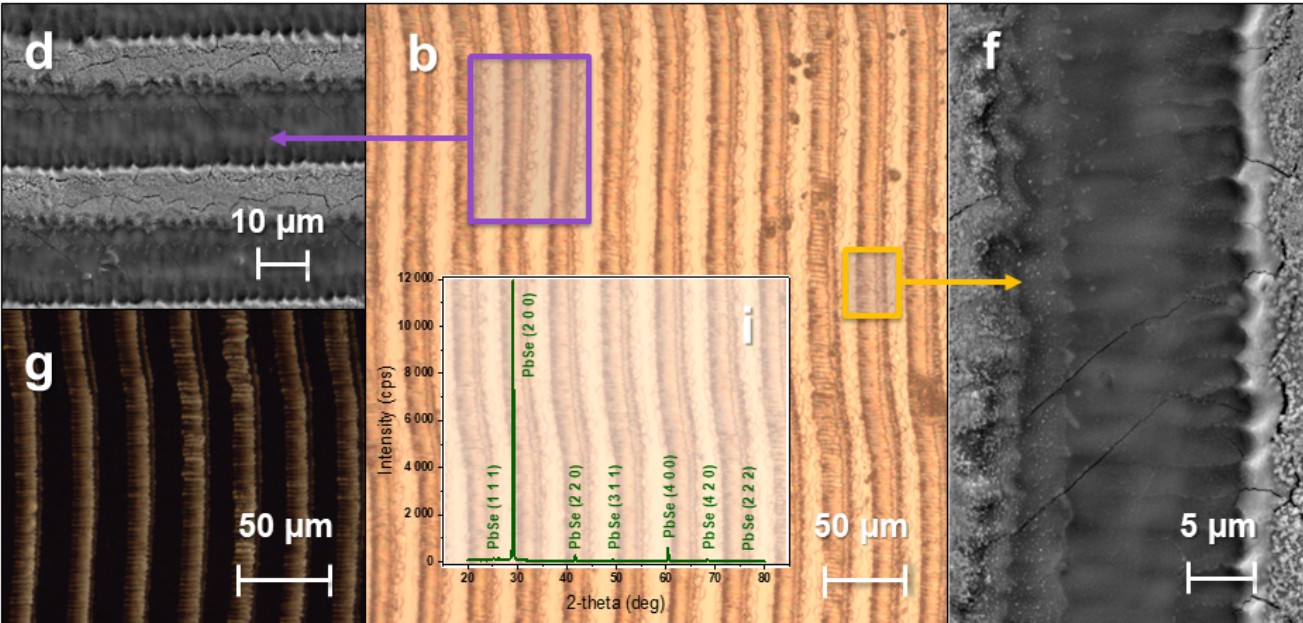

**Figure 5.** Images of a PbSe film obtained using optical microscopy in a bright field of reflected light: (**a**) darkening mode when modified by continuous radiation; (**b**) bleaching mode when modified by continuous radiation; in the field of transmitted light (**g**), the bleaching regime when modified by continuous radiation. Pictures of the PbSe film obtained with the SEM secondary electron detector: (**c**,**e**) darkening mode when modified by continuous radiation; (**d**,**f**) bleaching mode when modified by continuous radiation. Pictures of the PbSe film obtained using an X-ray diffractometer: (**h**) darkening mode when modified by continuous radiation; (**i**) bleaching mode when modified by continuous radiation.

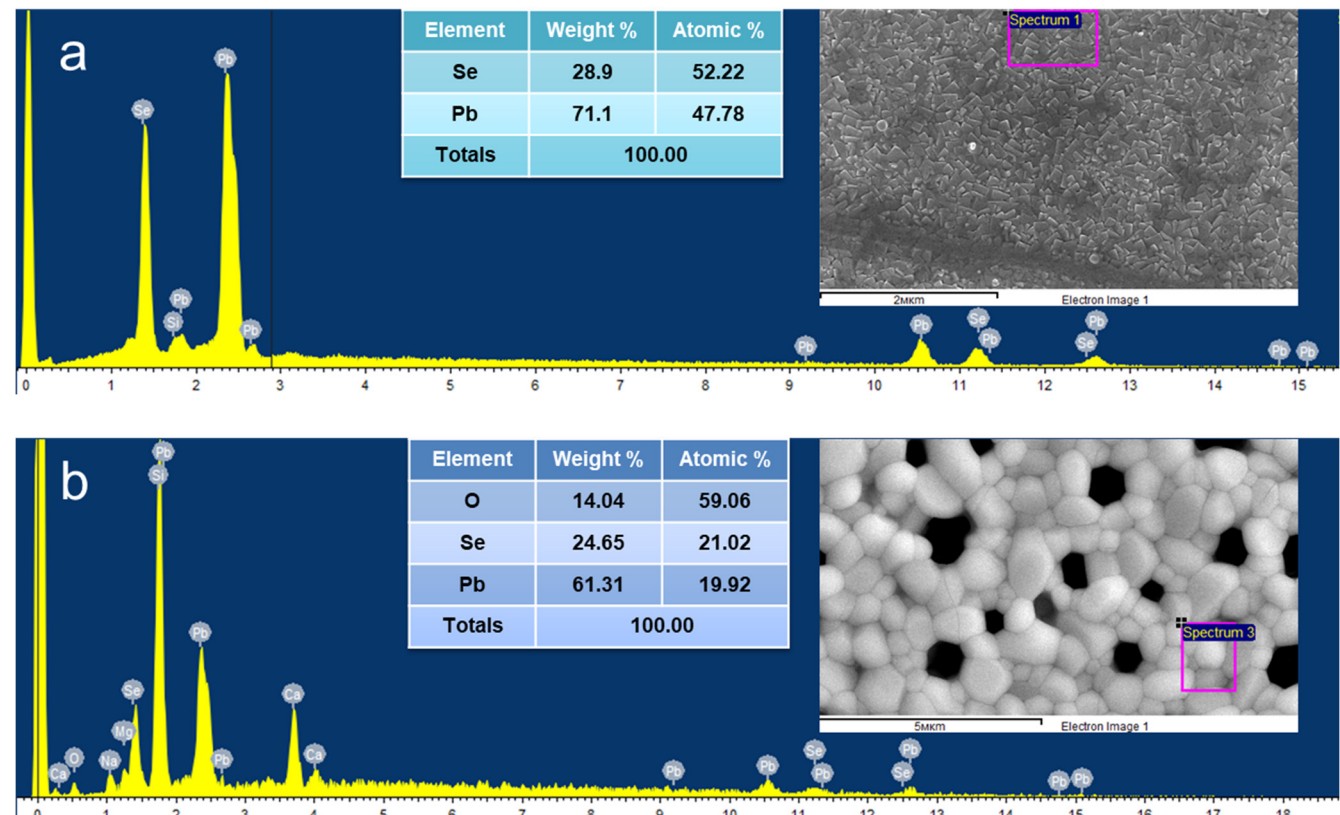

**Figure 6.** FESEM-EDX image: (**a**) original film, (**b**) film after heat treatment.

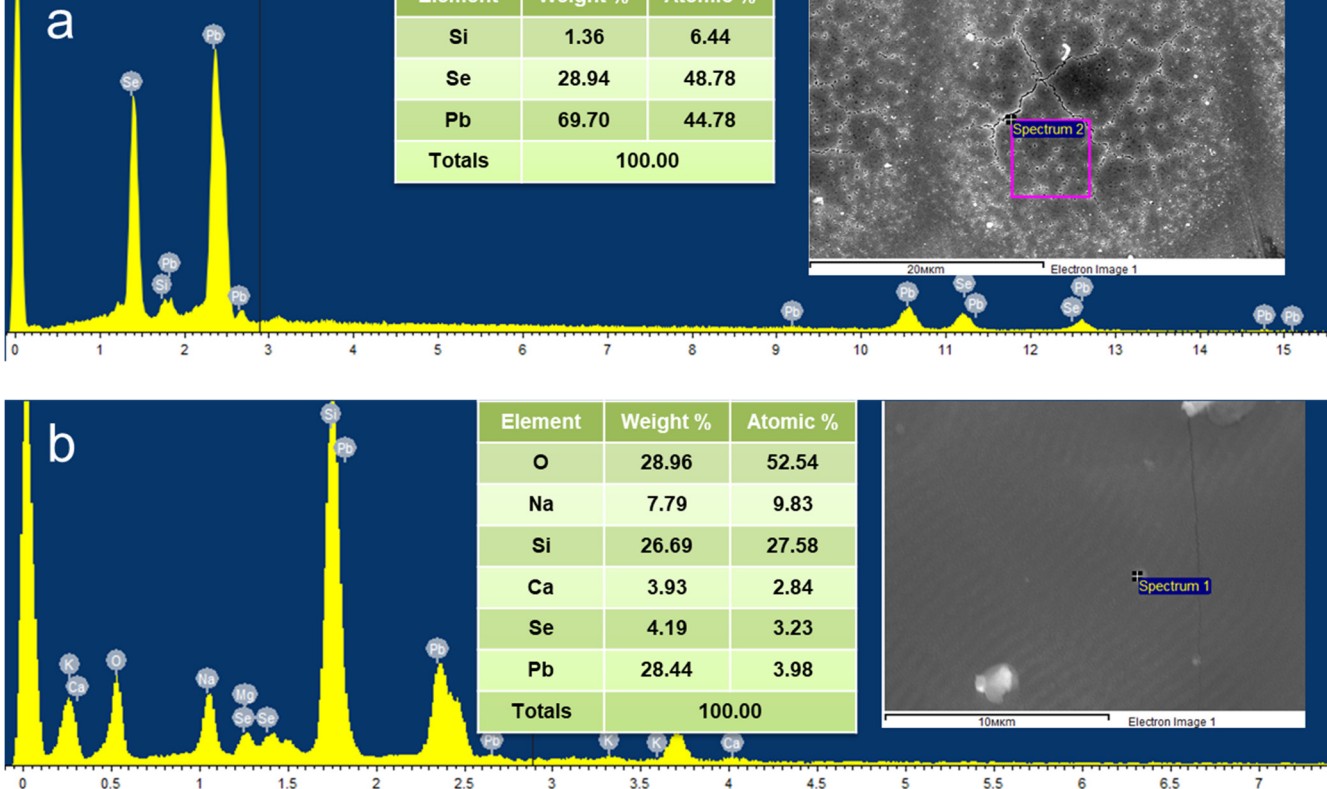

**Figure 7.** FESEM-EDX image: (**a**) darkening mode with pulsed radiation modification, (**b**) bleaching mode with pulsed radiation modification.

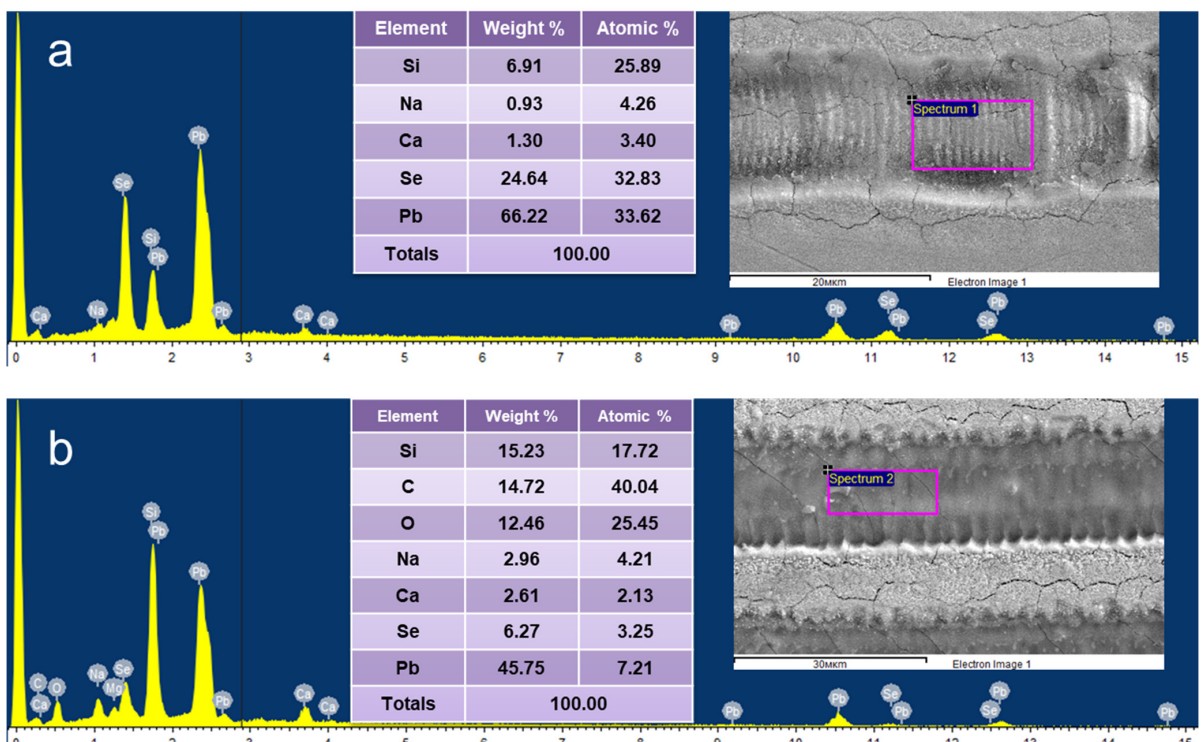

**Figure 8.** FESEM-EDX image: (**a**) darkening mode with CW modification, (**b**) bleaching mode with CW modification.

Figure 9 shows the reflectance spectra for samples that have undergone various processing types. The smallest reflection, almost over the entire measured wavelength range, was observed for the samples subjected to thermal treatment in the furnace, which is associated with the formation of a surface-transparent oxide. For PbSe films modified in sections using short laser pulses in the darkening mode, a sharp decrease in reflection was observed relative to the initial film but more so relative to heat treatment. However, in the exposure case to continuous laser radiation at a wavelength of 405 nm in the darkening mode, the reflection was less important than during heat treatment, and the modified regions approached a black body in their optical properties [23].

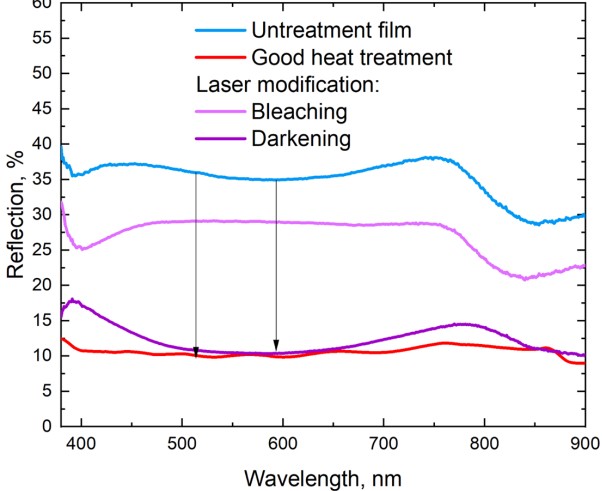

**Figure 9.** Reflectance spectra obtained for samples subjected to various treatments: untreated sample (blue curve), sample after heat treatment (red curve), sample after laser modification with a wavelength of 1064 nm in darkening mode (pink curve), and a sample after laser modification with a wavelength of 1064 nm in the bleaching mode (violet curve).

Below are the Fourier reflection and transmission spectra of the films for various processing methods (Figure 10).

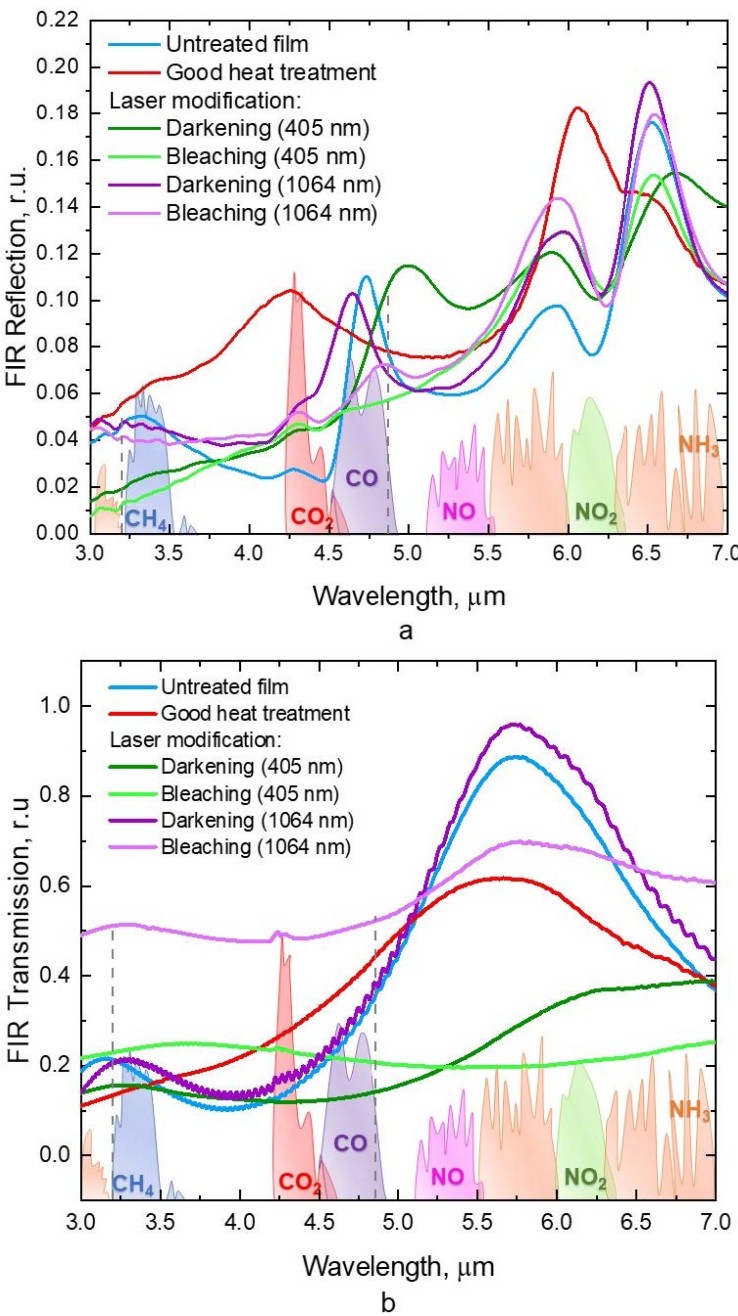

**Figure 10.** Fourier spectra of (**a**) reflection (excluding the film substrate) and (**b**) transmission (taking into account the film substrate) obtained for samples subjected to various treatments: initial sample (blue curve), sample after heat treatment (red curve). Sample after laser modification in the darkening mode modified by continuous radiation (dark green curve), a sample after laser modification in the bleaching mode modified by continuous radiation (light green curve), a sample after laser modification in the darkening mode modified by pulsed radiation (violet curve), a sample after laser modification in the bleaching mode modified by pulsed radiation (light violet curve). Absorption spectra of various gases ($CH_4$, $CO_2$, $CO$, $NO$, $NO_2$, $NH_3$).

As a result of heat treatment in the oven, an increase in reflection was observed (Figure 10a) relative to the initial PbSe film in the wavelength range from 3 μm to 4.5 and from 5 μm to 6.2 μm, which may be due to a change in the sample structure, including

the oxide surface formation. Laser-treated films have the least reflection, especially films treated with continuous radiation in the bleaching mode, in the wavelength range from 3 μm to 3.7 μm and from 4.6 μm to 4.8 μm, in which there are peaks of toxic gas absorption, $CH_4$ and CO, respectively [24]. The Fourier transmission spectra (Figure 10b) show that laser treatment at a wavelength of 405 nm in the darkening mode caused a decrease in transmission in the wavelength range from 4.2 μm to 5.2 μm. In the bleaching mode with modification by continuous radiation, in the range from 5.2 μm to 7 μm, the transmission of the sample also decreased relative to the initial film, and the optical characteristics of such a film approached an absolutely dark body. The film's transmission value after modification with continuous radiation was predominantly lower than the film's transmission value processed in the furnace, especially in the wavelength ranges corresponding to the absorption peaks of $CH_4$ and CO gases. During samples' laser processing using short pulses, it was observed that, in the bleaching regime, the films have a high transmission both in the bleaching and darkening regimes.

### 3.2. Calculation of Optical Constants

The measured characteristics were further used to determine the refractive index n, the extinction coefficient $k$, the absorption coefficient $\alpha$, and the band gap Eg [25]. The absorption coefficient was calculated by the expression: $\alpha = A/d$, where A is absorption and d is the film thickness (d $\approx$ 1 μm). The extinction coefficient was calculated in terms of $\alpha$ using the expression: $k = \alpha\lambda/(4\pi)$, where $\lambda$ is the wavelength. The ratio of reflected light from the film is expressed in terms of the Fresnel equations.

The value of $n$ was calculated in terms of light reflectance $R$ from the following equation [26]:

$$n = [(1 + R) + \{(1 + R)^2 - (1 - R)^2(1 + k^2)\}^{0.5}]/(1 - R)$$

where $R$—the reflective index.

This expression is applicable to the case when the film is strongly absorbing and the substrate influence can be neglected.

The expression was derived as a solution to the quadratic equation [26]:

$$R = [(n - 1)^2 + k^2/(n + 1)^2 + k^2]$$

Figure 11 shows the calculated optical constants of the untreated film and the film after laser modification. As a result of film laser modification, a decrease in the refractive index and an increase in the absorption coefficient and extinction coefficient were observed. Since the greatest increase in absorption was observed after laser treatment in the darkening mode, the extinction and absorption coefficients, which characterize the material absorbance, have the highest values for samples after laser modification during darkening. As a result of the photothermal action on the film with a high temperature gradient, the valence boundaries and conduction bands shifted, which affected the optical characteristics modification discussed above.

The PbSe film band gap was also determined before and after laser and heat treatment. By extrapolation of two straight sections, the following values were determined: untreated sample Eg = 1.59 eV, after laser modification on wavelength 1064 nm in the bleaching mode Eg = 1.45 eV, after heat treatment Eg = 1.33 eV, and after laser modification in the darkening mode Eg = 1.30 eV (Figure 12). In the authors' previous work [23], PbSe films, after modification by continuous radiation with a wavelength = 405 nm, had a band gap value in the darkening mode of Eg = 1.33. Since the Eg shift is fairly easy to achieve, some research groups [27,28] shift the band gap from intrinsic 0.27 eV to 1.8 eV, which corresponds to wavelengths of 4.6 μm and 690 nm, respectively. Thus, the band gap values obtained are in good agreement with the previously published results [25,28].

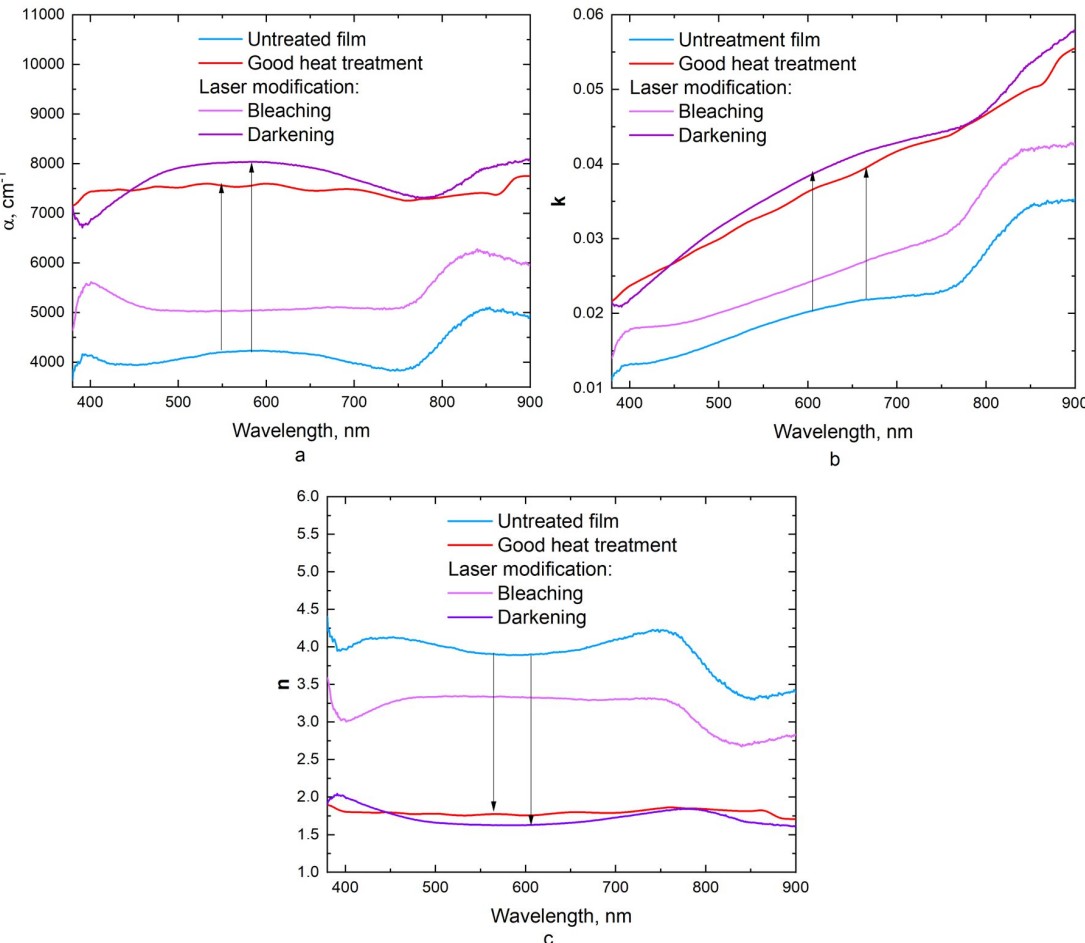

**Figure 11.** Optical characteristics of PbSe films for an untreated film (blue curve), a film after heat treatment (red curve), and a film after laser modification by pulsed radiation (pink and violet curves): (**a**) absorption coefficient α, (**b**) extinction coefficient k, (**c**) refractive index n.

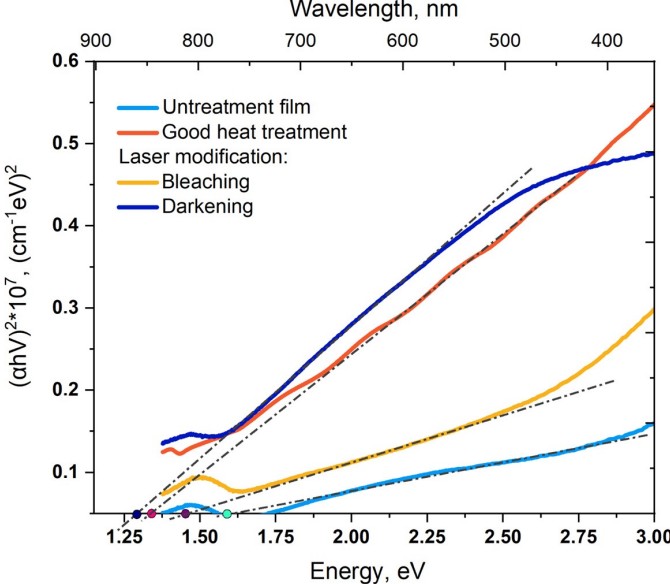

**Figure 12.** Determination of the optical band gap Eg before and after modification of PbSe films: untreated (light blue curve), heat treatment (red curve), bleaching mode (yellow curve), darkening mode (blue curve).

Structural changes caused by laser action led to an increase in the defect center concentration, which increased the probability of free charge carrier transfer. As a result, a decrease in the band gap was observed after laser modification [29].

Comparison of the advantages and disadvantages of processing with pulsed and continuous laser radiation with heat treatment is given in Table 3.

**Table 3.** Comparison of PbSe film processing methods.

| | | Advantages | Disadvantages |
|---|---|---|---|
| Heat treatment | | A long-established method Allows you to achieve increased absorption | Destructive influence Uncontrollable Growth of crystalline oxide Badly reproduced |
| NUV radiation | Darkening | The highest absorption is achieved in the visible range and in the range of 3–3.5 μm (CH$_4$ absorption peak) | Possible film degradation over time |
| | Bleaching | In contrast to heat treatment, a non-crystalline oxide is formed Achieves the highest absorption in the range 4.5–5 μm (CO absorption peak) | Possible film degradation over time Oxide formation on the surface |
| NIR radiation | Darkening | Allows you to achieve increased absorption | Possibly destructive effect |
| | Bleaching | Less destructive mode than darkening mode In contrast to heat treatment, a non-crystalline oxide is formed | Possibly destructive effect Oxide formation on the surface |

## 4. Conclusions

In the presented work, methods of modifying the optical properties of PbSe films were compared: laser modification in the bleaching mode and in the darkening mode using continuous radiation with a wavelength of 405 nm; laser modification using pulsed radiation with a wavelength of 1064 nm; and heat treatment in an oven.

The mode of laser exposure that makes it possible to provide the maximum decrease in reflection and the maximum absorption coefficient value, $\alpha = 9220 \text{ cm}^{-1}$ [25], was observed during scanning with continuous UV radiation in the darkening mode.

As a result of the laser exposure and treatment in the furnace, the refractive index n decreased, reaching its minimum value in the sample modification case in the darkening mode using short pulses. Moreover, this modification mode was characterized by a maximum decrease in the band gap value, Eg = 1.3 eV, relative to the raw film, which has Eg = 1.59 eV. However, the band gap value for samples treated in the bleaching mode using short laser pulses was 1.45 eV, which exceeded the value of Eg = 1.33 eV for films after thermal exposure in an oven and after laser exposure at a wavelength of 405 nm in the darkening mode.

The pulsed IR radiation used for modification turned out to be less effective, since the decrease in reflection is commensurate with modification upon annealing in a furnace.

The next stage of this study is to measure the electrical lead selenide films' characteristics after treatment with laser-pulsed IR radiation and compare with the results after exposure to UV radiation.

The results obtained show that film modification is a promising direction for improving optical characteristics. In the future, it will be possible to use processed chalcogenide films as organic substances sensors, as well as in gas analysis applications.

**Author Contributions:** Conceptualization, A.A.O. and M.M.S.; methodology, A.A.O., N.K.K., N.V.N. and M.A.D.; software, A.A.P. and M.A.D.; validation, A.A.O., M.A.D., A.A.P. and M.M.S.; formal analysis, A.A.P., N.K.K. and M.A.D.; investigation, A.A.P. and M.A.D.; resources, A.A.P. and M.A.D.; data curation, A.A.O.; writing—original draft preparation, A.A.O., A.A.P. and M.A.D.; writing—review and editing, M.M.S. and A.A.O.; visualization, A.A.O., N.K.K., N.V.N.; supervision, M.M.S. and A.A.O.; project administration, A.A.O.; funding acquisition, M.M.S. All authors have read and agreed to the published version of the manuscript.

**Funding:** This research was funded by the Russian Science Foundation grant (project No. 19-79-10208).

**Institutional Review Board Statement:** Not applicable.

**Informed Consent Statement:** Not applicable.

**Conflicts of Interest:** The authors declare no conflict of interest.

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
