# Peer review of "Comparison of CW NUV and Pulse NIR Laser Influence on PbSe Films Photosensitivity"

_applsci, doi:10.3390/app13042396_

Round 1

Reviewer 1 Report

The authors present a detailed study of the transformation of the optical properties of PbSe films upon excitation with continuous wave (near UV) laser and pulsed (IR) nanosecond laser pulses. They compare the laser treated films with the case of heat treatment in an oven. Although not clearly mentioned in the manuscript, the difference of using the radiation treatment is the possibility of producing bleaching and darkening of the sample. In the case of CW excitation, a clear advantage is the suppression of oxidation on the surface.

The study is quite complete and deserves to be published. In particular, the analysis of the optical properties is well performed and clear. However, the text needs major revision, since many aspects remain unclear for non-specialists.

For instance:

1) how are the darkening- and bleaching modes defined?

2) In which range of intensities for CW and pulsed radiation one obtains bleaching or darkening?

3) How is it possible to control bleaching and darkening?

4) What is the advantage of bleaching with subsequent destruction?

5) Is pulsed radiation better than heat treatment? If yes, why?

There are also some minor points:

6) There are some sentences throughout the manuscript which are unclear and need cross check using some translator.

7) In some figures the description of the curves is misleading (example Fig. 12, where the caption describes two “blue curves”).

In summary, I recommend publication after author revise the text and make the manuscript more readable for non-specialists.

Author Response

The authors present a detailed study of the transformation of the optical properties of PbSe films upon excitation with continuous wave (near UV) laser and pulsed (IR) nanosecond laser pulses. They compare the laser treated films with the case of heat treatment in an oven. Although not clearly mentioned in the manuscript, the difference of using the radiation treatment is the possibility of producing bleaching and darkening of the sample. In the case of CW excitation, a clear advantage is the suppression of oxidation on the surface.

The study is quite complete and deserves to be published. In particular, the analysis of the optical properties is well performed and clear. However, the text needs major revision, since many aspects remain unclear for non-specialists.

___________

We thank the reviewers for a thorough review of the manuscript, as well as for the comments and suggestions provided, which made it possible to qualitatively improve the text of the work and make the material presented more accessible to the general reader!

1) how are the darkening- and bleaching modes defined?

Answer: Thank you for asking this question. These modes are determined visually with optical microscopy. Also, the darkening mode is characterized by a decrease in reflection with a constant transmission. In the bleaching mode, an increase in transmission and a decrease in reflection can be observed.

2) In which range of intensities for CW and pulsed radiation one obtains bleaching or darkening?

Answer: We agree with the reviewer's remark so we have added tables with laser processing modes to help the reader better understand how the modes differ. We have added the following piece of text to the result and discussion.

«The parameters of laser processing, in which the effects of bleaching and darkening were observed, are shown in Tables 1, 2.

Table 1. Scanning modes with continuous NUV radiation.

Darkening

Bleaching

Scanning speed,  mm/s

1

1

Power density, kW/cm2

0.74

1.09

Table 2. Scanning modes with pulsed NIR radiation.

Darkening

Bleaching

Scanning speed, mm/s

120

13

Pulse frequency, kHz

5

50

Pulse duration, ns

4

20

Power density, kW/cm2

1.45

0.39

Despite the higher radiation power density in the darkening mode when exposed to NIR radiation, the energy density of the incident radiation is higher in the case of bleaching due to the lower scanning speed.

3) How is it possible to control bleaching and darkening?

Answer: Thank you for asking this question. It is possible to control bleaching and darkening by changing the energy density of laser radiation. Thus, in the work of the author, darkening manifested itself at a lower energy density of laser radiation, and bleaching at a higher one. In turn, the radiation energy density can be varied due to processing parameters, such as the laser spot diameter, pulse frequency, scanning speed, etc.

4) What is the advantage of bleaching with subsequent destruction?

Answer: Post-degradation of the film is an undesirable effect and has no particular advantage. However, as a result of laser treatment of the films in the antireflection mode, there is an increase in absorption in the visible spectral range and a decrease in reflection in the range of the CO absorption peak compared to the initial sample.

5) Is pulsed radiation better than heat treatment? If yes, why?

Answer: We agree with the reviewer's remark so a comparison of heat treatment with pulsed and continuous laser radiation was given.

We have added the following piece of text to the result and discussion.

«Comparison of the advantages and disadvantages of processing with pulsed and continuous laser radiation with heat treatment is given in Table 3.

Table 3. Comparison of PbSe film processing methods. »

Advantages

Disadvantages

Heat treatment

A long-established method

Allows you to achieve increased absorption

Destructive influence

Uncontrollable

Growth of crystalline oxide

Badly reproduced

NUV radiation

Darkening

The highest absorption is achieved in the visible range and in the range of 3 - 3.5 µm (CH4 absorption peak)

Possible film degradation over time

Bleaching

In contrast to heat treatment, a non-crystalline oxide is formed

Achieves the highest absorption in the range

4.5 – 5 µm (CO absorption peak)

Possible film degradation over time

Oxide formation on the surface

NIR radiation

Darkening

Allows you to achieve increased absorption

Possibly destructive effect

Bleaching

Less destructive mode than darkening mode

In contrast to heat treatment, a non-crystalline oxide is formed

Possibly destructive effect

Oxide formation on the surface

6) There are some sentences throughout the manuscript which are unclear and need cross check using some translator.

Answer: Thank you, we took into account your comment and tried to make the text of the article clearer.

7) In some figures the description of the curves is misleading (example Fig. 12, where the caption describes two “blue curves”).

Answer: We agree with the reviewer's remark so the description has been changed in Fig.12.

«Figure 12. Determination of the optical band gap Eg before and after modification of PbSe films: untreated (light blue curve), heat treatment (red curve), bleaching mode (yellow curve), darkening mode (blue curve).»

Reviewer 2 Report

The paper is devoted to laser processing of PbSe films in order to improve their photosensitivity for gas analysis issues. The content of the paper corresponds to the subject of the journal. But there are some comments to the paper:

- Why are lines selected as the scanning pattern ? If a pattern consisting of single laser spots is used as an array, the processing area can be significantly increased. It is necessary to justify the chosen solution.

- Do the mechanical properties of the film change after laser treatment? How does the adhesion of the film to the substrate change after laser treatment?

Author Response

We thank the reviewers for a thorough review of the manuscript, as well as for the comments and suggestions provided, which made it possible to qualitatively improve the text of the work and make the material presented more accessible to the general reader!

The paper is devoted to laser processing of PbSe films in order to improve their photosensitivity for gas analysis issues. The content of the paper corresponds to the subject of the journal. But there are some comments to the paper:

- Why are lines selected as the scanning pattern ? If a pattern consisting of single laser spots is used as an array, the processing area can be significantly increased. It is necessary to justify the chosen solution.

Answer: Thanks for the question. In this case, our choice of scanning pattern was chosen based on the previous article https://doi.org/10.3390/app121910162 to continue the study, deepen it and compare different radiation sources. We also relied on the previous work of our colleagues from the laboratory: Varlamov P. et. al // Materials Proceedings, 4(1), 1-6 (2021). However, we think the reviewer's idea is interesting and we will try to implement it in the following articles.

- Do the mechanical properties of the film change after laser treatment? How does the adhesion of the film to the substrate change after laser treatment?

Answer: Thanks for the question. To date, we have not conducted such a study. We thank the reviewer for important comments on the work for further development of the study and supplementing it with new information.

Reviewer 3 Report

In the manuscript entitled “Comparison of CW NUV and pulsed NIR laser influence on PbSe films photosensitivity” the authors have modified the PbSe thin films structure by using lasers. Structural and optical characteristics are studied in fresh and modified films. Although this study seems to be interesting, there are several flaws which need to be clarified. In general, I few a bit confusing about what experiments were exactly performed in this work. Which is darkening and bleaching mode? They are related to film with and without heat treatment? Also, what is the relation with gases peak wavelength range?  In my opinion, to accepted this manuscript, I recommend a major revision. The authors have state better the purpose of this work and clarify the realized measurements , in which film were laser processing were performed, etc.

Some minor localized errors also need to be done like:

References need to be uniformed.

Change coma to point in the numbers in several cases.

Fig. 3 Correct “…secondary electron detector: (c,e) initial film, (d,f) film after…”

Fig. 4 Correct “…secondary electron detector: (c,e) darkening mode …, (d,f) bleaching mode when…”

Fig. 5 Correct “…secondary electron detector: (c,e) darkening mode …, (d,f) bleaching mode when…”

Author Response

We thank the reviewers for a thorough review of the manuscript, as well as for the comments and suggestions provided, which made it possible to qualitatively improve the text of the work and make the material presented more accessible to the general reader!

In the manuscript entitled “Comparison of CW NUV and pulsed NIR laser influence on PbSe films photosensitivity” the authors have modified the PbSe thin films structure by using lasers. Structural and optical characteristics are studied in fresh and modified films. Although this study seems to be interesting, there are several flaws which need to be clarified. In general, I few a bit confusing about what experiments were exactly performed in this work.

Answer: The work is aimed at comparing the results of measurements of PbSe films using two laser sources. The impact of CW near ultraviolet (NUV) laser radiation was studied by us earlier in the article: Olkhova A. A., Patrikeeva A. A., Sergeev M. M. Electrical and optical properties of laser-induced structural modifications in PbSe films //Applied Sciences. - 2022. - T. 12. - No. 19. - P. 10162, in this article, the study was finalized. New results include the modification of the structure of PbSe films using near-infrared (NIR) laser pulses. Also, the results of the study of XRD and EDX of these samples were not previously presented.

Which is darkening and bleaching mode?

Answer: Visually, the two resulting modes: darkening and bleaching are different. In one case, the film becomes dark, in the second case, lighter. The modes under which the modification took place are partially presented in the article: https://doi.org/10.3390/app121910162.

They are related to film with and without heat treatment?

Answer: These modes are not related to heat treatment. Laser modification was carried out on the original films. The results of laser modification and heat treatment are shown together for comparison and a possible conclusion on the applicability of laser modification to increase the photosensitivity of detectors to replace heat treatment method.

Also, what is the relation with gases peak wavelength range? 

Answer: Chalcogenide films are excellent gas detectors because they have high absorption in the IR range, where gas molecules have absorption peaks. In this article, the peaks of gases are given for the reader's understanding of the approximate location of the peaks of hazardous gases.

In my opinion, to accepted this manuscript, I recommend a major revision. The authors have state better the purpose of this work and clarify the realized measurements, in which film were laser processing were performed, etc.

Answer: We thank the reviewer for the comment, we have added new data to the article, and also reflected in detail information about modification methods in the "Material and Methods" section:

«The original films that underwent laser modification were not subjected to heat treatment before. This study is aimed at comparing several processing modes of the original film: heat treatment, laser modification with different radiation sources, in order to identify the most photosensitive samples.»

Some minor localized errors also need to be done like:

References need to be uniformed.

Answer: We agree with the reviewer's remark so we have corrected these in article.

Change coma to point in the numbers in several cases.

Answer: We agree with the reviewer's remark so we have corrected these in article.

Fig. 3 Correct “…secondary electron detector: (c,e) initial film, (d,f) film after…”

Fig. 4 Correct “…secondary electron detector: (c,e) darkening mode …, (d,f) bleaching mode when…”

Fig. 5 Correct “…secondary electron detector: (c,e) darkening mode …, (d,f) bleaching mode when…”

Answer: We agree with the reviewer's remark so we have corrected these in article.

Reviewer 4 Report

In this paper, the surface modification of PbSe films by pulsed and continuous lasers was carried out to change the refractive index and hence the optical properties of light transmission, absorption and refractive index, and the results were compared with those obtained by heating in a furnace. The surface microstructure and the elemental composition of the surface were analyzed and the optical properties such as transmittance and reflectance were tested. However, the logic, the English presentation and the image processing of the paper still need further improvement. The main questions in the article are listed as follows:

1. According to the images under the microscope, periodic microstructures are evident on surfaces that have undergone continuous as well as pulsed machining. The periodic microstructure of the surface has an effect on the optical properties of the surface, which the authors would better take it into consideration.

2. The authors have carried out a surface elemental analysis of the machined surface. On the surface after laser processing, the surface elemental test results show elements such as Si, Na, Ca, etc. These should be the composition of the substrate glass, why are they detected through the EDX of the surface, is it because the surface is cracked after processing or for other reasons? The author would do well to clarify it.

3. In the Calculation of optical constants section, the meaning of R is not clearly stated in the formula for calculating n. Does R mean the reflective index?

4. In addition to laser processing there are also results obtained after heating in a furnace in the article, the author would be well advised to summarize and compare the advantages and disadvantages between the two methods.

5.In Figures 3, 4 and 5, the letters in the pictures do not seem to match the explanations below the picture, so the author needs to check and correct it carefully.

6. There are many literatures on the on the ultrafast laser-material interactions. The author should compare its advantages and disadvantages with others, such as:

[1] Study of the dynamics of material removal processes in combined pulse laser drilling of alumina ceramic. Optics & Laser Technology, 2023, 160:109053.

[2] Multi-scan picosecond laser welding of non-optical contact soda lime glass, Optics & Laser Technology, 2023, 161:109164.

[3] Experimental investigation on a new hybrid laser process for surface structuring by vapor pressure on Ti6Al4V, J. Mater. Process.Technol. 2020, 277:116450.

[4] Enhancing precision in fs-laser material processing by simultaneous spatial and temporal focusing, Light-Sci.Appl.2014, 3(5):e169.

Based on these remarks, I accept this manuscript for publication after major revision.

Author Response

We thank the reviewers for a thorough review of the manuscript, as well as for the comments and suggestions provided, which made it possible to qualitatively improve the text of the work and make the material presented more accessible to the general reader!

Reply to review:

  1. According to the images under the microscope, periodic microstructures are evident on surfaces that have undergone continuous as well as pulsed machining. The periodic microstructure of the surface has an effect on the optical properties of the surface, which the authors would better take it into consideration.

Answer: We agree with the reviewer's remark, when analyzing the images obtained using optical microscopy, we also found periodic microstructures in samples exposed to laser radiation. The authors suggest that such structures belong to laser-induced periodic surface structures (LIPSS). Colleagues in our laboratory are studying these structures [*], in the following works we will additionally take into account the influence of the resulting LIPSS on the optical properties of our films. However, in our previously published work [24], these structures did not appear on the surface, so the authors assume that they did not have a significant effect on the optical properties of the films.

[*] Sinev D. A. et al. Formation of the submicron oxidative LIPSS on thin titanium films during nanosecond laser recording //Nanomaterials. – 2020. – Т. 10. – â„–. 11. – С. 2161.

[24] Olkhova A. A., Patrikeeva A. A., Sergeev M. M. Electrical and Optical Properties of Laser-Induced Structural Modifications in PbSe Films //Applied Sciences. – 2022. – Т. 12. – â„–. 19. – С. 10162.

  1. The authors have carried out a surface elemental analysis of the machined surface. On the surface after laser processing, the surface elemental test results show elements such as Si, Na, Ca, etc. These should be the composition of the substrate glass, why are they detected through the EDX of the surface, is it because the surface is cracked after processing or for other reasons? The author would do well to clarify it.

Answer: Thanks for the comment. The following clarifications were added to the article:

“Сhemical elements, except Pb, Se and in some cases O, that are elements such as Si, Na, Ca, etc (Fig. 7-8) belong to the elementary composition of the cover glass-the film substrate. The reason for the detection of these elements is the destruction, including the cracking of the film under the action of laser radiation and the further removal of these elements to the surface of the samples”

  1. In the Calculation of optical constants section, the meaning of R is not clearly stated in the formula for calculating n. Does R mean the reflective index?

Answer: Yes, you are right, thank you for the comment. The authors added the designation R to the text: “where R is the reflective index”

  1. In addition to laser processing there are also results obtained after heating in a furnace in the article, the author would be well advised to summarize and compare the advantages and disadvantages between the two methods.

Answer: Thank you for drawing our attention. We have added a comparison table of three processing methods: heat treatment in a oven, treatment with pulsed and continuous laser radiation:

“Table 3. Comparison of PbSe film processing methods”.

  1. In Figures 3, 4 and 5, the letters in the pictures do not seem to match the explanations below the picture, so the author needs to check and correct it carefully.

Answer: Thank you very much, we found our mistake and fixed it. The corrected explanations to Figures 3, 4 and 5 are given below.

Figure 3. Images of a PbSe film obtained using optical microscopy in bright field reflected light: (a) initial film, (b) film after heat treatment.

Pictures of the PbSe film obtained with the SEM secondary electron detector: (c, e) initial film, (d, f) film after heat treatment.

A picture of the PbSe film obtained using an X-ray diffractometer (g) initial film, (h) after heat treatment.

Figure 4. Images of a PbSe film obtained using optical microscopy: in a bright field of reflected light (a) darkening mode when modified by pulsed radiation, (b) bleaching mode when modified by pulsed radiation; in the transmitted light field (g) the bleaching regime upon modification by pulsed radiation.

Pictures of the PbSe film obtained with the SEM secondary electron detector: (c, e) darkening mode when modified by pulsed radiation, (d, f) bleaching mode when modified by pulsed radiation.

Pictures of the PbSe film obtained using an X-ray diffractometer (h) darkening mode when modified by pulsed radiation, (i) bleaching mode when modified by pulsed radiation.

Figure 5. Images of a PbSe film obtained using optical microscopy in a bright field of reflected light: (a) darkening mode when modified by continuous radiation, (b) bleaching mode when modified by continuous radiation; in the field of transmitted light (g) the bleaching regime when modified by continuous radiation.

Pictures of the PbSe film obtained with the SEM secondary electron detector: (c, e) darkening mode when modified by continuous radiation, (d, f) bleaching mode when modified by continuous radiation.

Pictures of the PbSe film obtained using an X-ray diffractometer (h) darkening mode when modified by continuous radiation, (i) bleaching mode when modified by continuous radiation.”

  1. There are many literatures on the on the ultrafast laser-material interactions. The author should compare its advantages and disadvantages with others, such as:

[1] Study of the dynamics of material removal processes in combined pulse laser drilling of alumina ceramic. Optics & Laser Technology, 2023, 160:109053.

[2] Multi-scan picosecond laser welding of non-optical contact soda lime glass, Optics & Laser Technology, 2023, 161:109164.

Answer: Thank you so much for the articles you recommended, we have reviewed them. We do not plan to stop at considering only two laser sources. In the future, we plan to conduct experiments to modify the optical and electrical properties of our PbSe films using other laser radiation sources (with pico- and femtosecond pulse durations) and compare the results obtained. We would like to publish the final study in future articles.

Round 2

Reviewer 3 Report

The authors have addressed and answered all the queries.

Author Response

We thank the reviewer for a thorough review of the manuscript, as well as for the comments and suggestions provided, which made it possible to qualitatively improve the text of the work and make the material presented more accessible to the general reader!

Reviewer 4 Report

In this form, we could not find the revised content. The authors should point out the manuscript where is revised.

Author Response

We thank the reviewers for a thorough review of the manuscript, as well as for the comments and suggestions provided, which made it possible to qualitatively improve the text of the work and make the material presented more accessible to the general reader!

We have added all the necessary information about edits below, indicating the numbers of the lines with edits.

Reply to review: In this form, we could not find the revised content. The authors should point out the manuscript where is revised.

2. The authors have carried out a surface elemental analysis of the machined surface. On the surface after laser processing, the surface elemental test results show elements such as Si, Na, Ca, etc. These should be the composition of the substrate glass, why are they detected through the EDX of the surface, is it because the surface is cracked after processing or for other reasons? The author would do well to clarify it.

Answer: Thanks for the comment. The following clarifications were added to the article (line 271-275):

“Сhemical elements, except Pb, Se and in some cases O, that are elements such as Si, Na, Ca, etc (Fig. 7-8) belong to the elementary composition of the cover glass-the film substrate. The reason for the detection of these elements is the destruction, including the cracking of the film under the action of laser radiation and the further removal of these elements to the surface of the samples

3. In the Calculation of optical constants section, the meaning of R is not clearly stated in the formula for calculating n. Does R mean the reflective index?

Answer: Yes, you are right, thank you for the comment. The authors added the designation R to the text: “where R is the reflective index” (line 333)

4. In addition to laser processing there are also results obtained after heating in a furnace in the article, the author would be well advised to summarize and compare the advantages and disadvantages between the two methods.

Answer: Thank you for drawing our attention. We have added a comparison table of three processing methods: heat treatment in a oven, treatment with pulsed and continuous laser radiation:

“Table 3. Comparison of PbSe film processing methods”. (line 372-373)

5. In Figures 3, 4 and 5, the letters in the pictures do not seem to match the explanations below the picture, so the author needs to check and correct it carefully.

Answer: Thank you very much, we found our mistake and fixed it. The corrected explanations to Figures 3, 4 and 5 are given below.

Figure 3. Images of a PbSe film obtained using optical microscopy in bright field reflected light: (a) initial film, (b) film after heat treatment.

Pictures of the PbSe film obtained with the SEM secondary electron detector: (c, e) initial film, (d, f) film after heat treatment.

A picture of the PbSe film obtained using an X-ray diffractometer (g) initial film, (h) after heat treatment. (line 127-132)

Figure 4. Images of a PbSe film obtained using optical microscopy: in a bright field of reflected light (a) darkening mode when modified by pulsed radiation, (b) bleaching mode when modified by pulsed radiation; in the transmitted light field (g) the bleaching regime upon modification by pulsed radiation.

Pictures of the PbSe film obtained with the SEM secondary electron detector: (c, e) darkening mode when modified by pulsed radiation, (d, f) bleaching mode when modified by pulsed radiation. 

Pictures of the PbSe film obtained using an X-ray diffractometer (h) darkening mode when modified by pulsed radiation, (i) bleaching mode when modified by pulsed radiation. (line 187-195)

Figure 5. Images of a PbSe film obtained using optical microscopy in a bright field of reflected light: (a) darkening mode when modified by continuous radiation, (b) bleaching mode when modified by continuous radiation; in the field of transmitted light (g) the bleaching regime when modified by continuous radiation.

Pictures of the PbSe film obtained with the SEM secondary electron detector: (c, e) darkening mode when modified by continuous radiation, (d, f) bleaching mode when modified by continuous radiation.

Pictures of the PbSe film obtained using an X-ray diffractometer (h) darkening mode when modified by continuous radiation, (i) bleaching mode when modified by continuous radiation.” (line 217-225)